# SipD and IpaD induce a cross-protection against *Shigella* and *Salmonella* infections

**Bakhos Jneid, Audrey Rouaix, Cécile Féraudet-Tarisse, Stéphanie Simon** [ID]*

Université Paris Saclay, CEA, INRAE, Département Médicaments et Technologies pour la Santé (DMTS), SPI, Gif-sur-Yvette, France

* stephanie.simon@cea.fr

**Data Availability Statement:** All relevant data are within the manuscript and its Supporting Information files.

**Funding:** Bakhos Jneid obtained a grant from the PhD program of the Commissariat à l'Energie

## Abstract

*Salmonella* and *Shigella* species are food- and water-borne pathogens that are responsible for enteric infections in both humans and animals and are still the major cause of morbidity and mortality in the emerging countries. The existence of multiple *Salmonella* and *Shigella* serotypes as well as the emergence of strains resistant to antibiotics require the development of broadly protective therapies. Those bacteria utilize a Type III Secretion System (T3SS), necessary for their pathogenicity. The structural proteins composing the T3SS are common to all virulent *Salmonella* and *Shigella* spp., particularly the needle-tip proteins SipD (*Salmonella*) and IpaD (*Shigella*). We investigated the immunogenicity and protective efficacy of SipD and IpaD administered by intranasal and intragastric routes, in a mouse model of *Salmonella enterica* serotype Typhimurium (*S*. Typhimurium) intestinal challenge. Robust IgG (in all immunization routes) and IgA (in intranasal and oral immunization routes) antibody responses were induced against both proteins. Mice immunized with SipD or IpaD were protected against lethal intestinal challenge with *S*. Typhimurium or *Shigella flexneri* (100 Lethal Dose 50%). We have shown that SipD and IpaD are able to induce a cross-protection in a murine model of infection by *Salmonella* and *Shigella*. We provide the first demonstration that *Salmonella* and *Shigella* T3SS SipD and IpaD are promising antigens for the development of a cross-protective *Salmonella*-*Shigella* vaccine. These results open the way to the development of cross-protective therapeutic molecules.

## Author summary

*Salmonella* and *Shigella* are responsible for gastrointestinal diseases and continue to remain a serious health hazard in South and South-East Asia and African countries, even more with the emergence of multi drug resistances. Developed vaccines are either not commercialized (for *Shigella*) or cover only a limited number of serotypes (for *Salmonella*). There is thus a crucial need to develop cross-protective therapies. By targeting proteins SipD and IpaD belonging respectively to the injectisome of *Salmonella* and *Shigella* and necessary to their virulence, we have shown that these proteins are able to induce immune response and a cross-protection in a murine model of infection by *Salmonella*

Atomique et aux Energies Alternatives. The funders had no role in study design, data collection and analysis, decision to publish, or preparation of the manuscript.

**Competing interests:** The authors have declared that no competing interests exist.

and *Shigella* despite relatively weak identity sequence (38%). Such a candidate vaccine offers promising perspectives to control *Salmonella* and *Shigella* diseases.

## Introduction

*Salmonella* and *Shigella* are GRAM-negative enteropathogenic bacteria belonging to the *Enterobacteriaceae* family [1,2]. Both are responsible for gastrointestinal diseases ranging from moderate to acute, depending on different factors (e.g pathogen species, ingested dose, or immune status of the host). However, they continue to remain a serious health hazard in South and South-East Asia and African countries [3–7], causing notably severe diarrhea in children under the age of five in sub-Saharan Africa and south Asia [8–10]. Other at-risk populations include military personnel deployed abroad [11–13], travelers and victims of bioterrorist attacks [14,15]. While *Salmonella* and *Shigella* consist of only few species (two for *Salmonella*: *S. enterica* and *S. bongori* and four for *Shigella*: *S. flexneri*, *S. sonnei*, *S. dysenteriae* and *S. boydii*), there are a multiplicity of subspecies [16–18] making difficult the development of broad range vaccines.

Currently, three types of *Salmonella* vaccines are licensed: all of them target *S. enterica* serovar Typhi and do not offer cross-protection against other *Salmonella* serovars, or against non-typhoidal *Salmonella*. The situation is even worse for *Shigella* for which no licensed vaccine is available despite long standing efforts. Hopefully, these efforts will pay off in a next future in regards to the clinical trials currently evaluated worldwide [19–21].

Vaccine strategies can be grouped into two fundamental approaches: live-attenuated vaccines and nonliving vaccines. Live attenuated vaccines are generally more efficient to stimulate the immune response but generally do not induce a broad coverage. Non-living vaccines encompass inactivated whole-organisms or purified recombinant subunits. While offering the safest protection, they suffer from lower immunogenicity and generally require supporting strategies to overcome this hurdle [22–25].

Active immune system stimulation induced by vaccination takes days to weeks to be effective and can only be used to prevent infections. Because T3SS is essential for virulence and is conserved among all pathogenic *Salmonella* and *Shigella* strains [26], T3SS proteins appear as ideal candidates for *Salmonella-Shigella* vaccine and immunotherapy development. Type 3 secretion systems (T3SSs) or injectisomes are bacterial macromolecular organelles that are involved in the pathogenesis of many important human, animal and plant diseases [27] Bacteria that have sustained long-standing close associations with eukaryotic hosts have evolved specific adaptations to survive and replicate in this environment. The study of these systems is leading to unique insights into not only organelle assembly and protein secretion but also mechanisms of symbiosis and pathogenesis [26]. Components of T3SSs are widely distributed in GRAM-negative pathogens and are well conserved with regard to their overall structure, architecture, and function. The T3SS needle of *Salmonella* and *Shigella* is built by the helical polymerization of several hundred subunits of a single small protein (PrgI and MxiH respectively). The needle-tip is formed by a pentameric hydrophilic protein complex (SipD and IpaD respectively) connecting the distal end of the needle to the membrane spanning translocon (SipB, SipC for *Salmonella* and IpaB, IpaC for *Shigella*) [28–31]. During infection, the bacteria receive an external signal from the host environment and begin to assemble coordinately the constituents of the secretion system [32,33] which ultimately lead to the injection of effectors and/or invasion of the targeted host cell by the bacterium [34–39]. Based on the literature and our results, the needle-tip proteins have proved to be immunogenic in mice and in humans,

able to elicit good humoral responses protective against salmonellosis and shigellosis [2,40–44]. Moreover the sequence identity between IpaD and SipD [45], led us to the hypothesis that those needle tip proteins might be suitable targets for the development of a cross *Shigella*/*Salmonella* protective immunity. With this aim, we examined the immunogenicity of the *Salmonella* (SipD) and *Shigella* (IpaD) proteins, administered alone by comparing intranasal and intragastric immunization routes in a mouse model. We provide the first demonstration that SipD-IpaD are both promising target antigens for a cross-protective *Salmonella-Shigella* vaccine.

## Materials and methods

### Ethics statement

Six- to 8-week-old female BALB/c mice were purchased from Janvier Labs, France and maintained in accordance with the French and European regulations on care and protection of laboratory animals (European Community [EC] Directive 86/609, French Law 2001–486, 6 June 2001) and with agreement of the ethical committee (CETEA) no. 15–055 delivered to S. Simon and agreement D-91-272-106 from the Veterinary Inspection Department of Essonne (France). Up to eight mice were kept in each cage and housed in a temperature-regulated-room and had free access to food and water. All animals experiments were performed to ameliorate suffering according to the guideline of the CETEA committee.

### Bacterial strains

The *Salmonella enterica* serovar Typhimurium (CIP 104474, Pasteur Institute collection) and *Shigella flexneri 2a* (generous gift of Dr A. Phalipon, Pasteur Institute) were used in this study. Bacteria were first grown at 37°C on agar plates (trypticase soy (TCS) containing 0.01% Congo red (Serva) *for S. flexneri 2a* and LB plates *for S*. Typhimurium). For infection, a colony (Congo red-positive for *S. flexneri* 2a) was picked for a 5ml overnight (O/N) culture at 37°C in LB medium, followed by a culture in the same medium with 1:100 of the first culture for 2 h under the same conditions.

### Reagents

Biotin N-hydroxysuccinimide ester and streptavidin were from Sigma-Aldrich. Goat anti-mouse IgG and IgM polyclonal antibodies were from Jackson ImmunoResearch. Sandwich ELISAs were performed with MaxiSorp 96-well microtiter plates (Nunc, Thermoscientific), and all reagents were diluted in Enzyme Immuno-Assay (EIA) buffer (0.1 M phosphate buffer [pH 7.4] containing 0.15 M NaCl, 0.1% bovine serum albumin [BSA], and 0.01% sodium azide). Plates coated with proteins were saturated in EIA buffer (18 h at 4°C) and washed with washing buffer (0.01 M potassium phosphate [pH 7.4] containing 0.05% Tween 20). AEBSF (serine protease inhibitor) was from Interchim. Spectra/Por dialysis membranes were from Spectrum Laboratories. Cholera Toxin and Luria Broth were from Sigma. PBS was from Gibco by Life Technologies.

### Recombinant SipD and IpaD production and immunizations

The sipd and ipad genes from respectively *S.* Typhimurium and *S. flexneri* were synthesized (Genecust) based on the published sequences of *Salmonella* strain CIP 104474 and of *Shigella* strain CIP 82.48T and cloned into *Nde*I/*Xho*I restriction sites of the IPTG inducible pET22b vector (Novagen), allowing insertion of a poly-histidine tag sequence at the 3′ end of the genes (Table 1).

**Table 1. Sequences of the primers used for the cloning of sipd and ipad genes.**

| gene | name | sequence | |
|------|------|----------|---|
| sipd | sipd_nde1 | 5'-TATACATATGCTTAATATTCAAAATTATTCCGC-3' | |
| | sipd_xho1 | 5'-CAATAGGCCTCGAGTCCTTGCAGGAAGCTTTTGGCGG-3' | |
| ipad | ipad_nde1 | 5'-TATACATATGAATATAACAACTCTGACTAATAGTATT-3' | |
| | ipad_xho1 | 5'-CAATAGGCCTCGAGCTTTACCTCTTTTTCAAATAGACA-3' | |

Whole proteins SipD and IpaD were expressed and purified by affinity chromatography (Ni-NTA) as described previously [46]. Protein concentrations were determined by measuring absorbance at 280 nm ($A_{280}$) using the NanoDrop Spectrophotometer and the purity was assessed by SDS PAGE (10–15% gradient Phast Gel, Phast system, GE Healthcare). Purified recombinant proteins were stored at -20˚C until use.

Six- to 8-week-old female BALB/c mice were used by groups of 15. For intranasal (IN) immunizations, mice were anesthetized with isoflurane delivered through a vaporizer. Mice were immunized intranasally or intragastrically (IG, with a canula) on days 0, 21 and 42 with 10 μg of SipD or IpaD in 20 μL of PBS (IN) or 300 μg in 200 μL of phosphate-buffered saline (PBS) (IG). The proteins admixed with 1.5 μg (IN) or 10 μg (IG) cholera toxin adjuvant, were incubated for 1 h in a shaker at room temperature before immunization. Mice that received only adjuvant and PBS were included as controls. Animals were monitored daily after immunizations.

## LD50 determination and challenge procedures

**LD50 determination.** 5 mL of preculture of *S*. Typhimurium or *S. flexneri 2a* was grown in 200 mL of LB at 37˚C with agitation (200 rpm) until $OD_{600\ nm}$ ~1. Bacteria were centrifuged at 2,000 *x g* for 15 min at 4˚C and pellets were resuspended in PBS. Serial dilutions were performed in sterile PBS and approximately $2 \times 10^2$ to $2 \times 10^8$ CFU of *S*. Typhimurium were administered intra-gastrically (200 μL) using a curved gavage needle, or $5 \times 10^5$ to $5 \times 10^{10}$ CFU of *S. flexneri 2a* were administered intra-nasally (20 μL) to 20–22 week-old female BALB/c mice (5 mice per group). The exact number of CFU of each challenge dose was recalculated by viable counts (plating serial dilutions on LB agar plates). Mice were monitored twice daily for 25 days. The 50% mouse lethal dose (LD 50) for the challenge strains was calculated by the method of Reed and Muench and determined to be ~$10^4$ CFU/mL for *S*. Typhimurium ($2 \times 10^2$ CFU/mouse) and ~ $5 \times 10^8$ CFU/mL for *S. flexneri 2a* ($10^7$ CFU/mouse), in agreement with previous publication using this strain [47].

**Challenge.** On day 84 after primary immunization, mice (N = 15 per group, including control group: mice immunized intranasally with PBS+ adjuvant) were challenged with 100 LD 50 of virulent *S*. Typhimurium (~ $10^6$ CFU/mL, 200 μL in sterile PBS) via the intragastric route or with 100 LD 50 of virulent *S. flexneri 2a* (~ $5.10^{10}$ CFU/mL, 20 μL in sterile PBS) via intranasal route. Mice were monitored twice daily for 21 days after the challenge and health status, weight and survival were recorded. Any mouse that lost more than 20% of its initial body weight or showed advanced signs of morbidity was euthanized and scored as a death.

## Enzyme immunoassays

**Labeling with biotin.** One hundred μg of MAb or recombinant protein (SipD or IpaD) in 400 μL borate buffer (0.1 M; pH 8.5) was incubated at a 1:20 molar ratio with biotin-N-hydroxysuccinimide ester dissolved in 6 μL of anhydrous dimethylformamide (DMF). The reaction was stopped after 30 min at RT by adding 100 μL of 1 M Tris-HCl (pH 8) for 30 min. Finally, 500 μL of EIA buffer was added and the preparation was stored frozen at -20˚C until use.

**Evaluation of polyclonal response.** Anti-SipD/IpaD antibodies were measured in sera of immunized mice or hybridoma culture supernatants using sandwich ELISA. Briefly, microtiter plates were coated with 100 μL of goat anti-mouse Ig(G+M) antibodies or with rat anti-mouse IgG1, IgG2a, IgG2b antibodies at 10 μg/mL (diluted in 50 mM potassium phosphate buffer) overnight (ON) at RT. Plates were then saturated ON at 4˚C with 300 μL/well of EIA buffer. After a washing cycle performed with the washing buffer, 100 μL/well of serial dilutions of mouse sera (from $10^{-2}$ to $10^{-5}$) were added in duplicate and incubated overnight at 4˚C. The plates were then washed 3 times before adding 100 μL/well of biotinylated recombinant SipD or IpaD proteins at 100 ng/mL. Unrelated biotinylated recombinant proteins sharing also an His-tag at their C-terminus were sometimes added as controls (PrgI for SipD immunized mice and MxiH for IpaD immunized mice). After 2 hours of incubation at RT followed by three washing cycles, 100 μL/well of acetylcholinesterase (AChE; EC 3.1.1.7)-labeled streptavidin (1 Ellman unit/mL) were added and incubated for 1 hour at RT. Finally, the plates were washed 3 times and the absorbance was measured at 414 nm after 45 min of reaction with 200 μL/well of Ellman's reagent [48]. Concentrations of Ig(G+M) antibodies were calculated by fitting a calibrated control curve with nonlinear regression and interpolation of absorbance values of test samples by two-phase decay analysis.

## Statistical analysis

Graph Pad Prism 5 was used for the graphics generation and statistical analyses. The survival rates were analyzed using a two-tailed Fisher's exact test. Statistical analyses were performed using the non-parametric Mann-Whitney test to compare antibody concentrations between groups. Data are presented as the mean ± standard errors SEM for 10 or 15 samples per group of mice. A P value < 0.05 was considered significant in all determinations.

## Results

### Immunizations with SipD or IpaD proteins induce Ig(G+M) antibody responses

The SipD and IpaD proteins used to immunize mice were produced in *E. coli* BL21 (2.3 mg/L and 3 mg/L of culture of SipD and IpaD, respectively). Purity of proteins was assessed by SDS-PAGE electrophoresis and Coomassie blue staining (S1 Fig).

Mice immunized by the intranasal (IN) or the intragastric (IG) route with IpaD (Fig 1A, S2A Fig and S3A Fig) or SipD (Fig 1B, S2B Fig and S3B Fig) developed antigen-specific humoral responses. Total Ig (G+M), were measured using an ELISA test (principle of the ELISA in S4 Fig). Whatever the routes of immunization, the specific antibody titers against IpaD were superior to those obtained with SipD (Figs 1 and S2 and Table 2), probably because of a better immunogenicity of IpaD, compared to SipD. This hypothesis is supported by the results obtained with intragastric immunizations with SipD for which the specific Ig (G+M) responses are more heterogeneous and much lower (two logs, 0.25 μg/mL) than those obtained for IpaD. IpaD-specific Ig(G+M) concentrations reached the highest values by the IG route (23 μg/mL measured at day 84, one month after the third immunization, see Table 2). For both immunization routes, serum Ig (G+M) antibodies to SipD were detected before those to IpaD (after the second immunization) even if the final titer after the third immunization was higher for IpaD (S2 Fig). It should be noted that for the majority of Ig(G+M) measurements (Table 2), the concentrations were below the sum of the concentrations obtained for the different IgG isotypes. This could be due to the antibodies used for the standard curve in the sandwich ELISA: a mixture of specific SipD and IpaD IgG1/IgG2a/IgG2b was used as a standard of

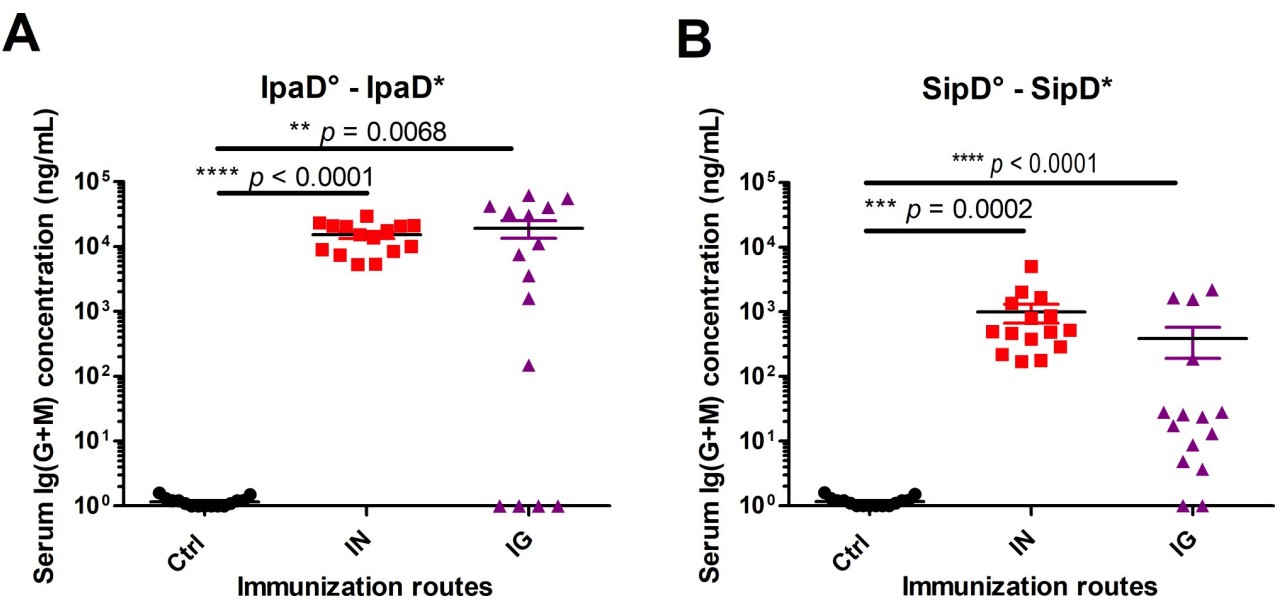

**Fig 1. Serum Ig(G+M) concentrations of mice immunized with IpaD or SipD.** Specific serum Ig (G+M) antibodies for IpaD (**A**) and SipD (**B**) were quantified by sandwich ELISA 2 weeks after the last immunization as described in experimental procedures. Data represent mean concentrations (ng/mL) and the standard errors (SEM) from 15 individual mice per group (control mice IN immunized with adjuvant + PBS). Asterisks and p values are indicated (**** $p < 0.0001$, *** $0.0001 < p < 0.001$, and ** $0.001 < p < 0.01$. Exact $p$ value indicated in the figure) when comparing mice immunized by the IN or IG route versus control mice using a nonparametric Mann-Whitney test.°: indicates injected immunogen; *: indicates biotinylated recombinant protein used for the ELISA analyses.

Ig(G+M) polyclonal antibodies, which does not exactly reflect the diversity of a polyclonal response (and particularly the IgM production), by comparison with the other tests where each specific isotype was used.

### Intranasal and intragastric administrations of SipD elicit serum IgA titers

To evaluate the induction of IgA antibodies by the mucosa, the first line of adaptive immune defense against enteric pathogens, IpaD and SipD specific IgA titers in serum from immunized and control mice were measured (Fig 2A and 2B respectively and Table 2). For each protein, the specific IgA titers were equivalent for mice immunized intranasally or intragastrically. It should be noted that for SipD some of the mice did not produce any detectable IgA, contrary to what was noted for IpaD, which supports what we observed for Ig(G+M) responses and the hypothesis of a better immunogenicity of IpaD protein.

**Table 2. Summary of the homologous (Ig (G+M), IgG1, IgG (2a+2b), IgA) and heterologous (Ig (G+M)) antibody responses after the last immunization with SipD or IpaD by the IN and IG routes.**

| Immunization route | Immunogen | Homologous antibody response | | | | Heterologous response |
| --- | --- | --- | --- | --- | --- | --- |
| | | Ig(G+M) | IgG1 | IgG(2a+2b) | IgA titer | Ig (G+M) |
| IN | SipD | $1.7 \times 10^3$ | $2.6 \times 10^3$ | $4.6 \times 10^2$ | $2.4 \times 10^2$ | $4.4 \times 10^1$ |
| | IpaD | $2.9 \times 10^4$ | $2.0 \times 10^4$ | $6.8 \times 10^3$ | $5.2 \times 10^2$ | $2.9 \times 10^1$ |
| IG | SipD | $1.2 \times 10^3$ | $6.8 \times 10^3$ | $9.9 \times 10^2$ | $1.7 \times 10^2$ | $4.4 \times 10^0$ |
| | IpaD | $2.6 \times 10^4$ | $4.9 \times 10^2$ | $8.3 \times 10^3$ | $3.8 \times 10^2$ | $1.9 \times 10^2$ |

Data represent mean concentrations (ng/mL) for Ig(G+M), IgG1, IgG(2a+2b) responses and IgA titer from each group of mice.

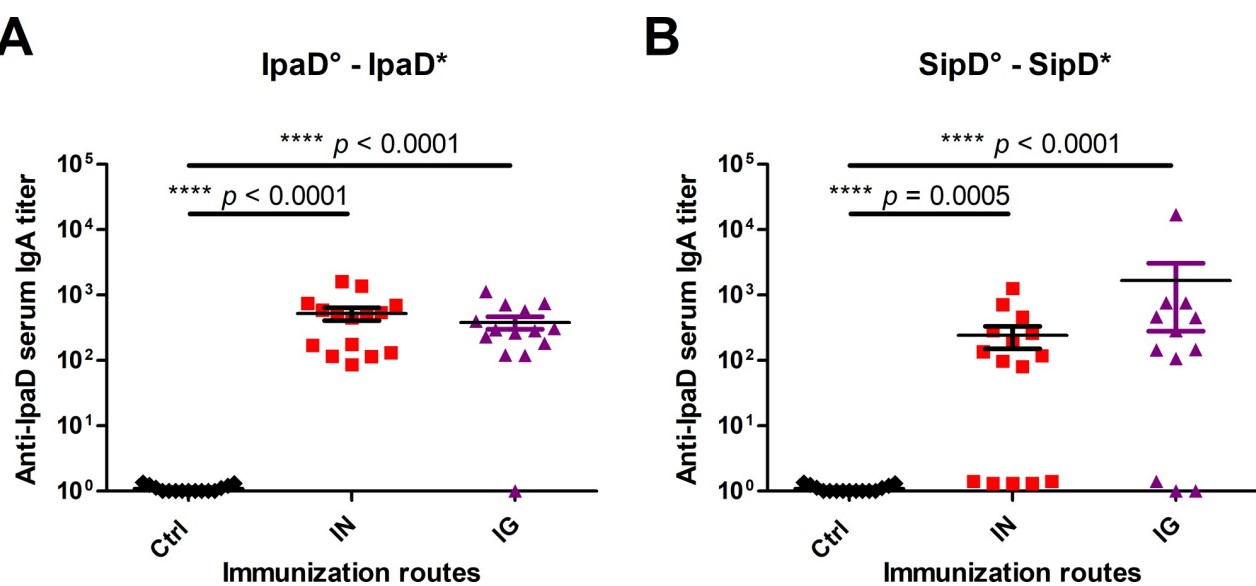

**Fig 2. IgA titers of mice immunized with IpaD or SipD.** Specific serum IgA antibody titers for IpaD (**A**) and SipD (**B**) were quantified by sandwich ELISA 2 weeks after the last immunization as described in experimental procedures. Data represent mean titers and the standard errors (SEM) from 15 individual mice per group (control mice IN immunized with adjuvant + PBS). Asterisks and p values are indicated (**** $p < 0.0001$ and *** $0.0001 < p < 0.001$, exact $p$ value indicated in the figure) when comparing mice immunized by the IN or IG route versus control mice using a nonparametric Mann-Whitney test.°: indicates injected immunogen; *: indicates biotinylated recombinant protein used for the ELISA analyses.

### Immune response involved all main IgG isotypes in serum

To investigate further the immune response elicited by both routes of immunization for both proteins, the IpaD and SipD homologous specific IgG1, IgG2a and IgG2b subclasses were measured in serum from immunized IpaD, SipD and control mice after the third immunization for the IN and IG routes (Fig 3 and Table 2). Measurement of the IgG isotype concentrations in sera of immunized mice revealed that all main subclasses contributed to the humoral response whatever the route. Anti-IpaD IgG1 were found in higher concentration after IN route immunization compared with IG route (Fig 3A, left panel), whereas for IgG (2a + 2b), the levels were equivalent (Fig 3A. right panel). For SipD, no difference was found between the two routes either for IgG1 or IgG (2a + 2b) (Fig 3B). It has to be mentioned that whatever the subtypes of anti-SipD immunoglobulins, concentrations were slightly inferior to the ones obtained for IpaD and responses were more heterogeneous, reflecting differences of immunogenicity of the two proteins. IgG1 and IgG(2a+2b) are respectively indicators of the T helper type 2 (humoral) and type 1 (cellular) immune responses. IgG (2a+2b):IgG1 ratios were taken as indicators of the T helper type 1 (Th1, cellular response)/Th2 (humoral response) balance, in order to evaluate the contribution of each pathway to the immune response. As *Salmonella* and *Shigella* are facultative intracellular pathogens and multiply in macrophages, one could expect the involvement of the cellular immunity during an infection. IpaD and SipD were able to induce a similar response by the IN route with a ratio close to 1 and slightly in favor of IgG1 production (humoral response) (Fig 4). The balance was more clearly in favor of a cellular response for IpaD by the IG route (ratio around 10), opposite to the result obtained for SipD for which a humoral immunity was favored (ratio IgG (2a+2b):IgG1 close to 0.1). However this result should be taken with caution as it has not been confirmed by measuring directly the T cell specific response.

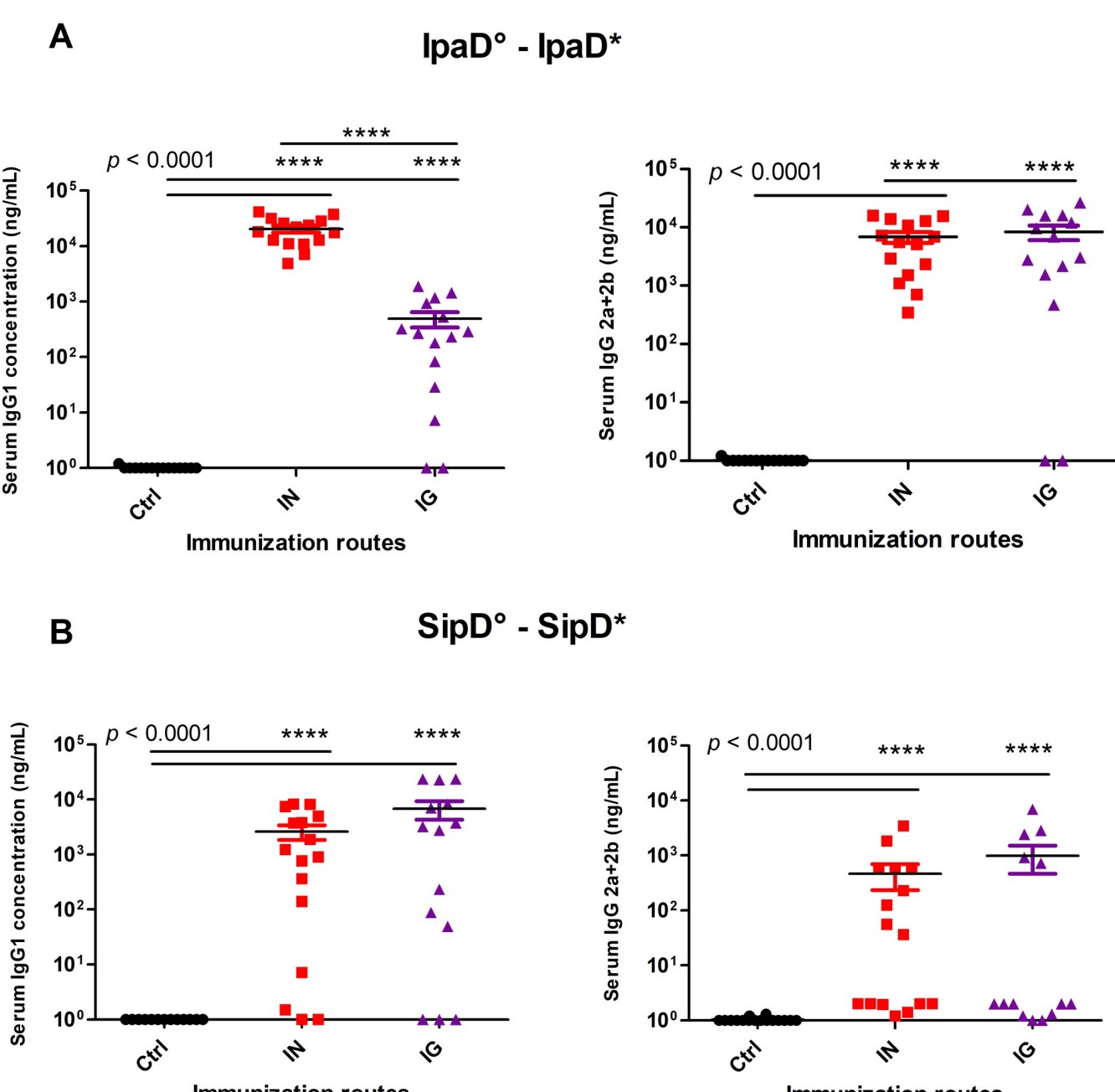

**Fig 3. Serum IgG subtype concentrations of mice immunized with IpaD or SipD.** Serum IgG1 (left panels), IgG2a and IgG2b (right panels) subclasses specific for IpaD **(A)** and SipD **(B)** were quantified by sandwich ELISA, 2 weeks after the last immunization. Data represent mean concentrations (ng/mL) and the standard errors (SEM) from 14–16 mice per group. Asterisks and p values are indicated (**** $p < 0.0001$, exact $p$ value indicated in the figure) when comparing IG or IN immunized mice versus control mice (control mice IN immunized with adjuvant + PBS), as well as IN vs IG routes for IpaD IgG1.°: indicates immunogen injected; *: indicates biotinylated recombinant protein used for the ELISA analyses.

## Heterologous antibody responses

Because the needle-tip proteins SipD and IpaD of the T3SS of *Salmonella* and *Shigella* present sequence identities and with the aim of studying the possibility of cross-protection, the humoral responses against SipD in mice immunized with IpaD and against IpaD in mice immunized with SipD were measured. The crossed (heterologous) Ig(G+M) antibody

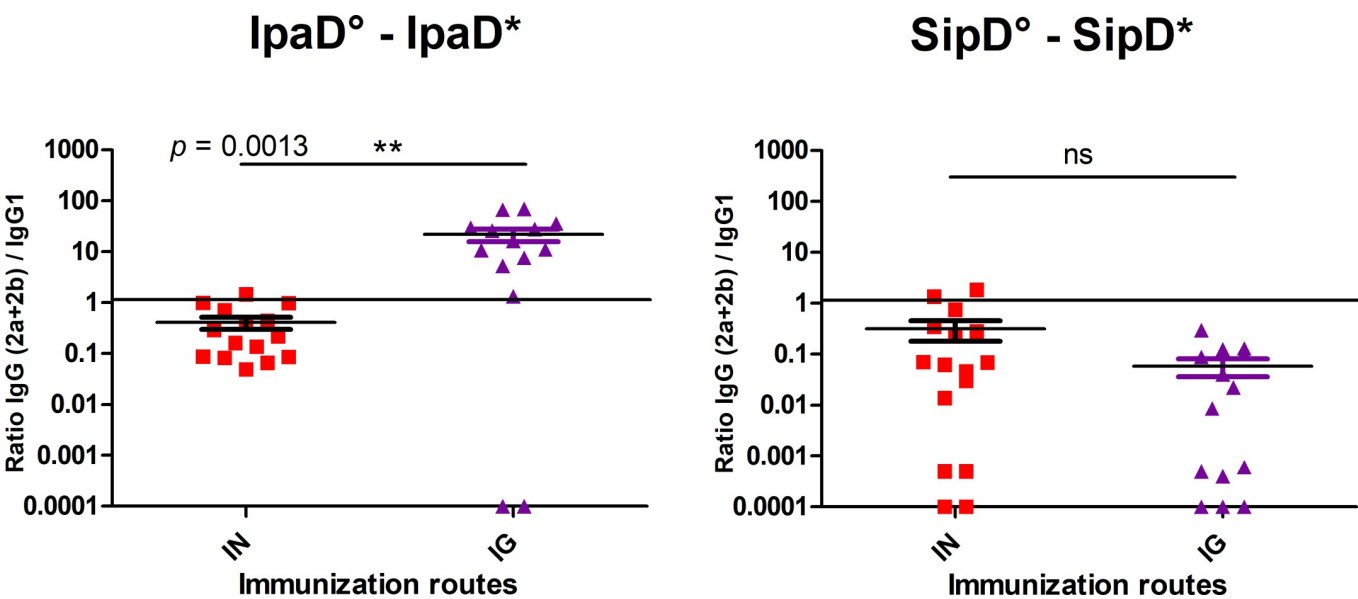

**Fig 4.** IgG (2a +2b) / IgG 1 ratio after IpaD (A) and SipD (B) immunizations. Data represent mean titers and the standard errors (SEM) from 15 individual mice per group. Asterisks and p values are indicated (** 0.001 < *p* < 0.01, ns: non-significant. Exact *p* value indicated in the figure) when comparing mice immunized by the IN or IG route using a nonparametric Mann-Whitney test.°: indicates injected immunogen; *: indicates biotinylated recombinant protein used for the ELISA analyses.

responses were significantly lower (approximately 100-fold) than the specific (homologous) responses (S5 Fig for the kinetics, compare Fig 5 for heterologous response to Fig 1. for the homologous one, and see Table 2). IpaD immunogen seems to induce a higher heterologous

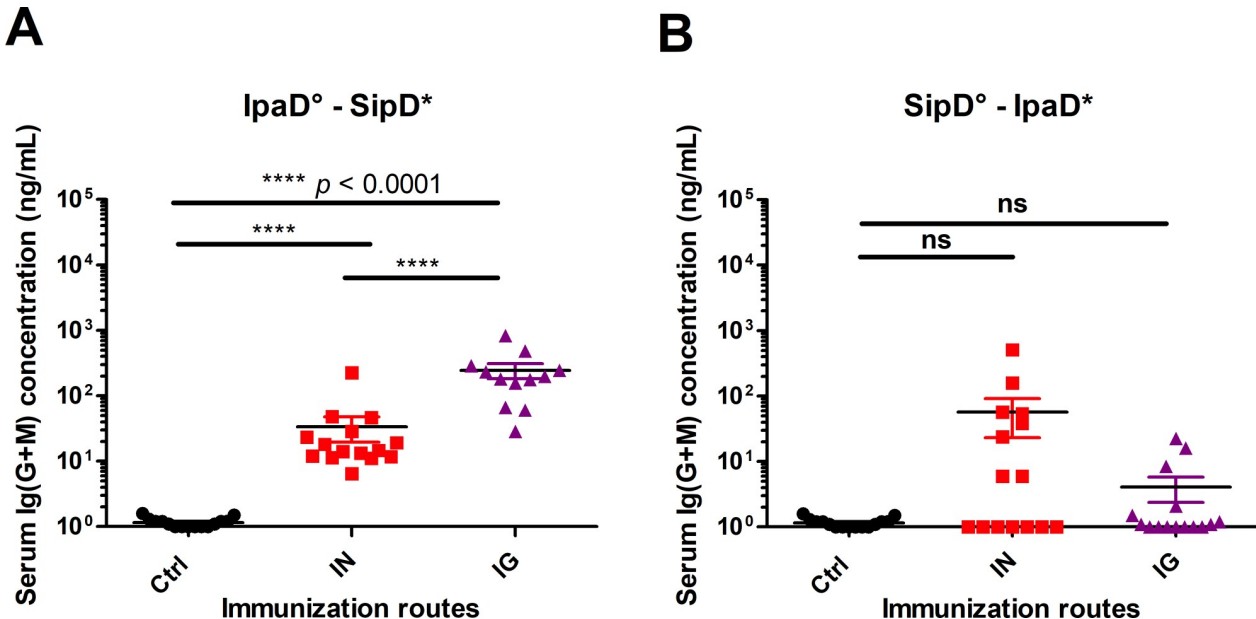

**Fig 5. Heterologous serum Ig(G+M) concentrations of mice immunized with IpaD or SipD.** Specific serum Ig (G+M) antibodies against SipD for mice immunized with IpaD (**A**) and against IpaD for mice immunized with SipD (**B**) were quantified by sandwich ELISA 2 weeks after the last immunization as described in experimental procedures. Data represent mean concentrations (ng/mL) and the standard errors (SEM) from 15 individual mice per group (control mice IN immunized with adjuvant + PBS). Asterisks and p values are indicated (**** *p* < 0.0001. Exact *p* value indicated in the figure. ns: non-significant) when comparing mice immunized by the IN or IG route versus control mice using a nonparametric Mann-Whitney test.°: indicates injected immunogen; *: indicates biotinylated recombinant protein used for the ELISA analyses.

antibody response against SipD than the opposite and particularly by the IG route (Table 2), confirming a better immunogenicity/stability of IpaD protein after administration or a better accessibility of the conserved regions between IpaD and SipD when IpaD is used as immunogen. These results are in agreement with the higher production of homologous anti-IpaD antibodies compared to homologous anti-SipD antibodies.

### Protective efficacy against lethal *S.* Typhimurium or *S. flexneri 2a* challenge

The lethal doses 50% (LD50) of the *S.* Typhimurium (intragastric infection) and *S. flexneri 2a* (intranasal infection) strains used in the experiments (see experimental procedures) were determined at $10^4$ CFU/mL for *Salmonella* ($2X10^2$ CFU/mouse) and $5.10^8$ CFU/mL for *Shigella* ($10^7$ CFU/mouse) (S6 Fig) according to the Reed and Muench method [49] which is in agreement other studies ([2,47,50]. To assess first the homologous protective efficacy induced by SipD against *S.* Typhimurium and IpaD against *S. flexneri 2a*, immunized and control mice were subjected to intragastric or intranasal challenge, six weeks after the last immunization, with a high dose of bacteria: ~ 100 LD50 of *S.* Typhimurium ($2X10^4$ CFU/mouse) or *S. flexneri 2a* ($10^9$ CFU/mouse) (Fig 6A–6C, respectively, and Table 3). In all challenges, the mortality rate of control animals (mice administered phosphate buffer saline [PBS]/adjuvant) was 100% with death occurring at 16–21 days after challenge by *S.* Typhimurium and at 8–13 days after challenge by *S. flexneri 2a*. SipD and IpaD were able to induce efficient homologous protection against challenge by their bacterial counterparts (Fig 6A–6C). The best homologous protective efficacy was induced by IpaD against a *S. flexneri 2a* challenge (intragastric route, 61% survival rate). In order to evaluate the cross-protective efficacy of each of the proteins, mice immunized intragastrically or intranasally with IpaD or SipD were challenged by *S.* Typhimurium and *S. flexneri 2a*, respectively (Fig 6B–6D, and Table 3). Weak cross-protection induced by IpaD was obtained against *S.* Typhimurium infection by the IN and IG routes (27% and 30%, respectively). Cross-protection induced by SipD against *S. flexneri 2a* challenge was significant and even superior to the homologous protection induced by IpaD, whatever the route of immunization (47% by IN route, 67% by IG route). We hypothesized that these cross-protections could be due to the production of specific antibodies directed against crucial regions common to both proteins, although their protein sequence identity is relatively weak (38%, S7 Fig).

Heterologous protection induced by SipD was equivalent to the homologous protection induced by IpaD against *S. flexneri 2a* infection (40% vs 27% by IN route, 60% vs 61.5% by IG route) while the Ig(G+M) antibody concentration able to cross-react with SipD in IpaD-immunized mice seemed to be higher than the one produced against IpaD in SipD-immunized mice. It has to be noted that cross-reactive IgA in SipD-immunized mice were not measured and could bring substantial protection against *S. flexneri 2a* infection.

## Discussion

Infections caused by *Shigella* and *Salmonella* (typhoidal as well as nontyphoidal, and particularly invasive nontyphoidal *Salmonella* (iNTS)) are associated with a high burden in terms of mortality and morbidity especially in low income countries and in children under 5 years of age [51]. For this reason, long-standing efforts have been made to understand the immunological mechanisms underlying these infections and to develop effective therapies against them. Vaccines targeting typhoidal *Salmonella* are already marketed, but none protect against nontyphoidal *Salmonella*. No licensed vaccine exists for *Shigella*, though some developments have been the subject of clinical studies with varying degrees of success. The existence of multiple *Shigella* and *Salmonella* serotypes and the increase of multiresistant iNTS as well as *Shigella* clones highlight the need for development of a broad-spectrum protective vaccine [52,53].

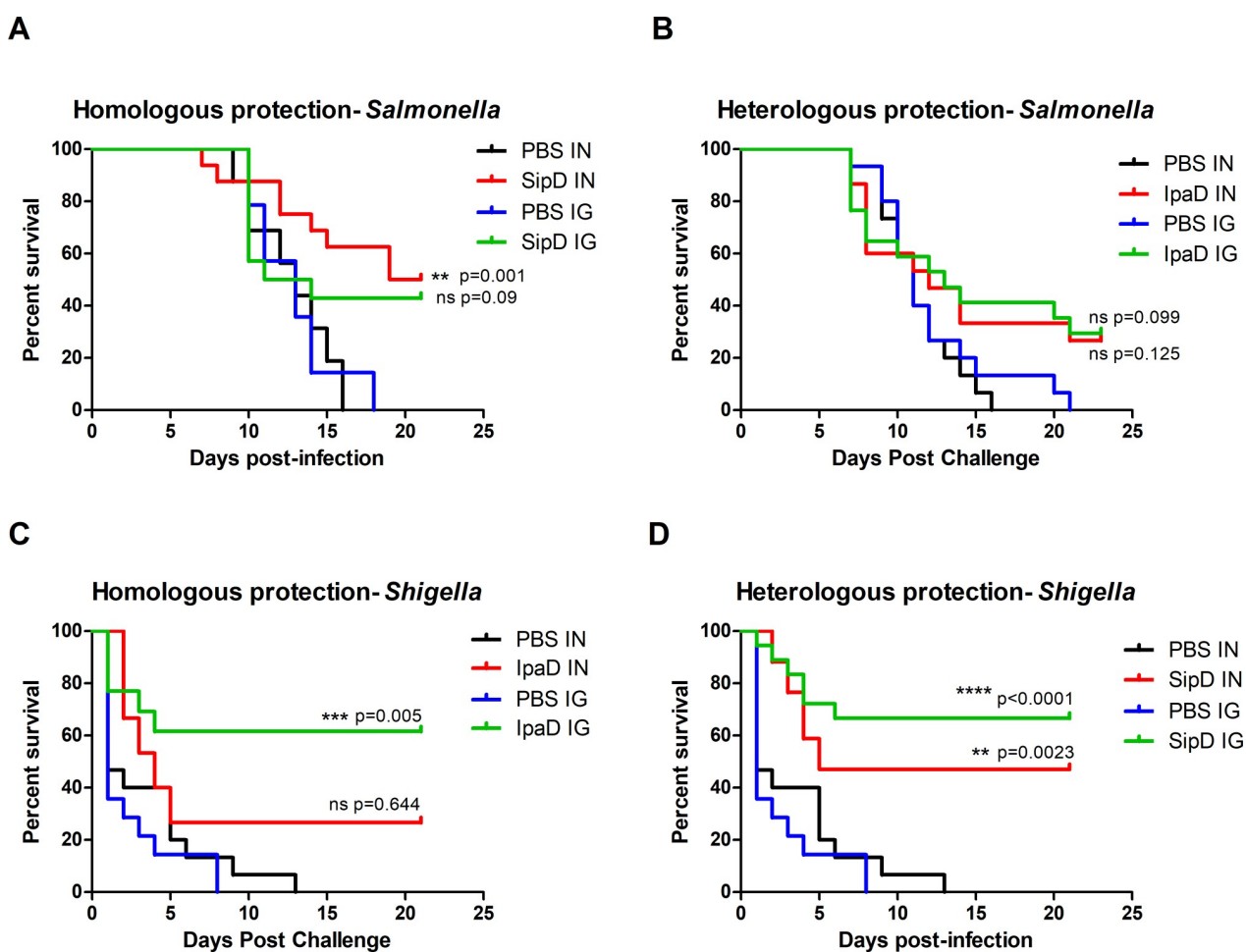

**Fig 6. Homologous and heterologous protective efficacies induced by SipD and IpaD immunizations against *S.* Typhimurium and *S. flexneri* 2a challenges.** Mice (N = 15) were immunized at days 0, 21 and 42 by the indicated antigens (or by adjuvant + PBS for controls) and routes. Six weeks after the last immunization, at day 84, 100 LD 50 of *S.* Typhimurium (**A, B**) or *S. flexneri 2a* (**C, D**) was administered (intragastrically and intranasally, respectively) to SipD (**A, D**) or IpaD (**B, C**) immunized mice. Survival was monitored for 21 days. Statistical significance was determined using a log-rank (Mantel-Cox) test. Statistically significant differences are indicated by **** $p < 0.0001$, *** $0.0001 < p < 0.001$, and ** $0.001 < p < 0.01$. Exact $p$ value indicated in the figure. ns: non-significant) compared to PBS groups.

Different studies show the importance of the humoral response in the fight against *Salmonella* and *Shigella* infections [54–60]. For *Shigella*, numerous data provide evidence of the immunogenicity/protective role of T3SS proteins and particularly IpaB/IpaD [42,59,61–64], which have been evaluated as parts of a bi-component recombinant vaccine [43]. Although it is recognized that the mouse model of *Shigella* pulmonary infection is not ideal to mimic an intestinal infection, it is currently the one used by scientific community for the evaluation of vaccines in development. For *Salmonella*, studies on the importance of a protective humoral response are scarcer and sometimes controversial ([65] and for review see [66]). We have shown in a preliminary study that SipD induced a good humoral response and was protective against a *S.* Typhimurium challenge [46], and more recently Martinez-Becerra and coll. have shown that two fusion proteins mixed together and composed of SipB/SipD and SseB/SseC have the potential to provide a cross-protective effect against two serovars of *Salmonella enterica* [67]. However, unlike for IpaB/IpaD of *Shigella*, the SipB/SipD fusion protein alone was unable to elicit protection.

**Table 3. Homologous and cross-protection efficacy induced by SipD and IpaD T3SS protein immunizations by the IN and IG routes from lethal challenge with *S. flexneri 2a* (intranasal) or *S.* Typhimurium (intragastric) in mice.**

| Immunization route | Immunogen | Challenge | Homologous protection efficacy (%) | Heterologous protection efficacy (%) | P value [a] |
|---|---|---|---|---|---|
| IN | IpaD | *S. flexneri 2a* | 27 | | 0.644 |
| | IpaD | *S.* Typhimurium | | 27 | 0.125 |
| | SipD | *S.* Typhimurium | 50 | | 0.001 |
| | SipD | *S. flexneri 2a* | | 47 | 0.002 |
| IG | IpaD | *S. flexneri 2a* | 62 | | 0.0005 |
| | IpaD | *S.* Typhimurium | | 30 | 0.099 |
| | SipD | *S.* Typhimurium | 43 | | 0.090 |
| | SipD | *S. flexneri 2a* | | 67 | <0.0001 |

The mice immunized by the intranasal (IN) or intragastric (IG) route with SipD and IpaD were challenged with $10^9$ CFU/mouse of *S. flexneri 2a* by the IN route (LD 50 = $10^7$ CFU/mouse) or with $2X10^4$ CFU/mouse of *S.* Typhimurium by the IG route (LD 50 = $2X10^2$ CFU/mouse). The mortality rate of the immunized group was compared with that of the PBS-immunized control animals using the log-rank (Mantel-Cox) test.

Based on these data and because of the sequence identity, the strong similarity in the three-dimensional structures and the mechanism of action between SipD and IpaD, as well as the role of the humoral response against these proteins and their importance in protecting against *Shigella* and *Salmonella* infections, we have hypothesized that broad-spectrum cross-protection against *Salmonella* and *Shigella* infections can be induced by using SipD or IpaD as immunogen. The results of this study show that by using indifferently SipD or IpaD, good protection (60%) against *Shigella flexneri 2a* infection is obtained despite very high challenging doses (100 LD50). In a comparative study, using the same model of *Shigella* pulmonary infection, immunizations with IpaD yielded 70 to 90% cross-protection against 5 and 11 LD50 of *S. sonnei* and *S. flexneri*, respectively, which decreased dramatically to around 20% with 9 and 24 LD50 of *S. sonnei* and *S. flexneri*, respectively [68]. Interestingly, we found in this study that protection against *S. flexneri* was equivalent using SipD or IpaD, whereas the immune responses induced by SipD were lower than those induced by IpaD. This difference might be due to better immunogenicity of IpaD compared to SipD. This hypothesis is supported by the results obtained with IG immunizations with SipD for which the specific Ig (G+M) responses are more heterogeneous and much lower (two logs) than those obtained for IpaD, which could be partly explained by the heterogeneity in the degradation of the proteins by the gastric acid of the stomach. In addition, while *Shigella* infections are carried out intranasally, protection is better when immunizations are performed intragastrically. This may be related, among other things, to a slightly higher IgA titer by the IG route than by the IN route, induced by immunizations with SipD. All antibody subclasses are involved in the humoral response for both SipD and IpaD, regardless of route of immunization, and this highlights the importance of humoral (and particularly mucosal) immunity in protecting against *Salmonella* and *Shigella* infection. It has to be noted that protection against *Shigella* infection is better when mice are immunized by the IG route compared to the IN route. This might be correlated to the IgG subtype measurement balance in favor of a cellular response for IpaD by the IG route suggesting a significant contribution of the Th1 response for protection after IN challenge. It has to be noted that this hypothesis has not been verified by a direct measurement of the T cell specific response.

Although IpaD is also able to induce protection against *Salmonella* infection (100 LD50), it is nevertheless lower than that obtained for *Shigella* with SipD. This might be due to different factors that could be linked altogether: i) *Salmonella* has two type three secretion systems involved in the pathogenicity [69], ii) pathogenicity mechanisms are different between *Salmonella* and *Shigella* and particularly in regard to the involvement of the innate and adaptative

immune response, and iii) a more important systemic dissemination of *Salmonella* in the murine model [70,71]. Nevertheless, the protective effect obtained using SipD/IpaD as immunogen underscores the importance of the extracellular life cycles of *Salmonella* and *Shigella* for their pathogenicity and dissemination and highlight the role of conserved regions of needle-tip proteins SipD/IpaD in this protection.

To our knowledge, a cross-protective effect of a T3SS-1 component against *Shigella* and *Salmonella* infections has never been described before this study. The novelty of the results obtained here should highlight the major role of SipD and IpaD in *Salmonella* and *Shigella* virulence and although give first evidence of the interest of these proteins as potential targets to protect broadly against *Salmonella* and *Shigella* infection in development of new vaccines. The role of SipD/IpaD effectors in systemic dissemination of these bacteria strengthens the protective effect obtained using these proteins as immunogens and underscores the importance of their extracellular life cycle for their pathogenicity and dissemination. The common molecular mechanisms governing the cross-protection induced by SipD or IpaD remain now to be deciphered. Because of the key role of SipD and IpaD in the virulence of the bacteria, they are well conserved among the different *Salmonella* and *Shigella* strains and species and thus appears as good targets for broad-spectrum coverage against different *Salmonella* and *Shigella* species and serotypes. However, further investigations are needed to evaluate this possibility.

## Supporting information

**S1 Fig. Analysis of recombinant SipD and IpaD proteins.** SDS-PAGE / Coomassie blue staining (reducing conditions) of purified recombinant proteins. PolyHis-IpaD (37.1 kDa, lane 2) and polyHis-SipD (38.2 kDa, lane 3) are shown with molecular mass markers in kilodaltons (kDa) (lane 1).
(TIF)

**S2 Fig. Kinetics of homologous polyclonal Ig(G+M) antibody responses to IpaD and SipD antigens.** Mice were immunized three times (time indicated with arrows) with IpaD **(A)** or SipD **(B)** by the IN route (left panels) or IG route (right panels) as described in Materials and Methods. Homologous responses of Ig(G+M) antibodies specific for IpaD or SipD were quantified by sandwich ELISA. Data represent mean concentrations (ng/mL) and the standard errors (SEM) from 14–16 individual mice per group. (**** $p < 0.0001$, *** $0.0001 < p < 0.001$, ** $0.001 < p < 0.01$ and * $0.01 < p < 0.1$. ns: non significant) comparing the antibody responses on days post-immunization versus those on day 0 (nonparametric Mann-Whitney test). °: indicates injected immunogen; *: indicates biotinylated recombinant protein.
(TIF)

**S3 Fig. Example of specificity of polyclonal Ig(G+M) antibody responses to IpaD and SipD antigens.** Mice were immunized three times intranasally (IN) or intragastrically (IG) with IpaD **(A)** or SipD **(B)** as described in Materials and Methods. Example of specificity of Ig(G+M) responses is shown for one mouse per route of immunization, and was assessed by using biotinylated unrelated recombinant proteins, sharing the same His-tag as IpaD and SipD at their C-terminus. Control (ctl) His-tagged MxiH (needle protein of *Shigella* injectisome) or His-tagged PrgI (needle protein of *Salmonella* injectisome) were used for mice immunized with IpaD and SipD respectively and quantified by sandwich ELISA. Data represent absorbance units obtained with sera of mice diluted 1000 fold.
(TIF)

**S4 Fig. Principle of sandwich ELISA used for measurement of circulating antibodies.** A sandwich ELISA test was performed to measure the concentrations of circulating antibodies

(immune response after immunizations (Ig(G+M), IgG1, IgG2a, IgG2b and IgA, see experimental procedures)
(TIF)

**S5 Fig. Kinetics of heterologous polyclonal Ig(G+M) antibody responses to IpaD and SipD antigens.** Mice were immunized three times (time indicated with arrows) with IpaD **(A)** or SipD **(B)** by the IN route (left panels) or IG route (right panels) as described in Materials and Methods. Heterologous responses of Ig(G+M) antibodies specific for SipD (from mice immunized with IpaD) or SipD (from mice immunized with IpaD) were quantified by sandwich ELISA. Data represent mean concentrations (ng/mL) and the standard errors (SEM) from 14–16 individual mice per group. (**** $p < 0.0001$, *** $0.0001 < p < 0.001$, ** $0.001 < p < 0.01$ and * $0.01 < p < 0.1$. ns: non significant) comparing the antibody responses on days post-immunization versus those on day 0 (nonparametric Mann-Whitney test). °: indicates injected immunogen; *: indicates biotinylated recombinant protein.
(TIF)

**S6 Fig. Determination of LD50 for S. Typhimurium and S. flexneri 2a.** Serial dilutions of *S.* Typhimurium (from $2.10^2$ to $2.10^8$ CFU) and *S. flexneri 2a* ($5.10^5$ to $5.10^{10}$ CFU) were administered intragastrically (*S.* Typhimurium) or intranasally (*S. flexneri 2a*) to 20- to 22-week-old female BALB/c mice (5 mice per group). The 50% mouse lethal dose (LD 50) was calculated by the method of Reed and Muench.
(TIF)

**S7 Fig. Alignment of IpaD and SipD sequences from S. flexneri 2a and S. Typhimurium.** Alignment sequences of IpaD from *S. flexneri 2a* (accession number SVF87366.1) and SipD from *S.* Typhimurium (accession number AAA86617.1) were performed using BLAST (Basic local alignment search tool) from NCBI (https://blast.ncbi.nlm.nih.gov/). SipD sequence is represented in blue and IpaD sequence in red. Identical aminoacids are represented in black and similar aminoacids by a "+". Sequence identity is 38.17%.
(TIF)

## Acknowledgments

We thank Dr Armelle Phalipon for the generous gift of the *Shigella flexneri 2a* strain.

## Author Contributions

**Conceptualization:** Bakhos Jneid, Stéphanie Simon.

**Formal analysis:** Bakhos Jneid, Stéphanie Simon.

**Funding acquisition:** Stéphanie Simon.

**Investigation:** Bakhos Jneid, Audrey Rouaix, Cécile Féraudet-Tarisse.

**Methodology:** Bakhos Jneid.

**Project administration:** Stéphanie Simon.

**Resources:** Stéphanie Simon.

**Supervision:** Stéphanie Simon.

**Validation:** Stéphanie Simon.

**Writing – original draft:** Bakhos Jneid, Stéphanie Simon.

**Writing – review & editing:** Stéphanie Simon.

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
