## [Decision Letter · Decision Letter 0]

24 Feb 2020

Dear Dr. Simon,

Thank you very much for submitting your manuscript "SipD and IpaD induce a cross-protection against Shigella and Salmonella infections" for consideration at PLOS Neglected Tropical Diseases. As with all papers reviewed by the journal, your manuscript was reviewed by members of the editorial board and by several independent reviewers. In light of the reviews (below this email), we would like to invite the resubmission of a significantly-revised version that takes into account the reviewers' comments. 

We cannot make any decision about publication until we have seen the revised manuscript and your response to the reviewers' comments. Your revised manuscript is also likely to be sent to reviewers for further evaluation.

Sincerely,

Alfredo G Torres

Deputy Editor

Alfredo Torres

Deputy Editor

Reviewer's Responses to Questions

**Key Review Criteria Required for Acceptance?**

**Methods**

-Are the objectives of the study clearly articulated with a clear testable hypothesis stated?

-Is the study design appropriate to address the stated objectives?

-Is the population clearly described and appropriate for the hypothesis being tested?

-Is the sample size sufficient to ensure adequate power to address the hypothesis being tested?

-Were correct statistical analysis used to support conclusions?

-Are there concerns about ethical or regulatory requirements being met?

Reviewer #1: The methods that are described are sufficient but it is unclear why the authors chose to measure Ig concentrations in ng/ml rather than the typical titers. Also, no discussion of PrgI production is presented making that protein irrelevant. 

The LD50 for Shigella is very, very high. There is no discussion of differential colony staining on Congo red. It is likely that the vast majority of colonies are white producing an artificially high LD50. At these challenge doses all the mice should be dead in 24 hr.

Reviewer #2: Methods have been done and presented properly.

Reviewer #3: This is a well written manuscript giving a reasoned account of how the authors selected the antigens to induce a cross-protection against Shigella and Salmonella infections in a mouse model. The experiment was well designed and the homologous and heterologous antibody responses following the selected antigens immunization makes this work interesting.

Reviewer #4: As seen in the comments to the authors, the following concerns are noted for the methods: 

- the analyses used histidine-tagged proteins or biotinylated the tagged proteins for the ELISA analyses

- no direct measure for T cell-specific responses.

- the LD50 for the Shigella infection resulted in an 80% survival and required a very high infectious dose for the challenge studies.

**Results**

-Does the analysis presented match the analysis plan?

-Are the results clearly and completely presented?

-Are the figures (Tables, Images) of sufficient quality for clarity?

Reviewer #1: The results are presented but for most of the manuscript are overstated. It is hard to the manuscript seriously when IpaD does not provide protection after IN vaccination and IN challenge. That should have been the positive control. Again, with such a high dose of Shigella, it is unclear how any of the mice could have survived. Like the shigella are "white" and are missing the T3SS.

The authors need to decide on OG vs IG. This is OG ORO-gastric

Reviewer #2: Like Methods, results have been done and presented properly. However, since the authors purified the recombinant proteins, I think it was better if they share some photos of the recombinant proteins.

Reviewer #3: The results are completely presented. However, the quality and resolution for Figures 1- 6 are not satisfied.

Reviewer #4: The analyses match the plan and the figures/tables are presented clearly. Please note the minor concerns to improve the clarity of the figure legends.

**Conclusions**

-Are the conclusions supported by the data presented?

-Are the limitations of analysis clearly described?

-Do the authors discuss how these data can be helpful to advance our understanding of the topic under study?

-Is public health relevance addressed?

Reviewer #1: The conclusions are NOT supported by the results. Many of the conclusions drawn by the authors lack significance - especially in the challenge experiments.

Reviewer #2: (No Response)

Reviewer #3: Conclusion is well-defined and supports the data presented. It is helpful to advance our understanding of the topic under study.

Reviewer #4: Based on the concerns noted in the methods and in the comments to the authors, some of the conclusions are not supported by the data presented. In particular, the cross-protection, efficacy in the Shigella challenge model, and the T helper cellular responses are not supported. The limitations to these are not clearly described.

**Editorial and Data Presentation Modifications?**

Reviewer #1: (No Response)

Reviewer #2: There were some little writing mistakes that I should mention to be corrected:

-In the first line of the abstract please write food- and water-borne pathogens

-In line 37, please write emergence of multi drug resistances.

-In the lane 74, what is 30? 

-Please italicize family, genus, species and subspecies all over the manuscript. (Line 48, for example). 

-Please capitalize the word "Gram" in line 79.

-In Lane 135, please write "phosphate-buffered saline". 

-I think there should be a space between the numbers (4, 37, etc.) and ºC. Please correct it all over the text.

-In line 142, there should be a space between g and for.

-In line 143 and 144, please write 2 × 102 to 2 × 108 CFU and 5 × 105 to 5 × 1010 CFU. 

-All over the text you have used capital L for showing the volume, but somewhere, mainly in "Enzyme immunoassays" part in Materials and Methods section. Please correct this.

-In line 167, there should be a space between 10 and μg/mL.

Reviewer #3: The manuscript is recommended for publication once the minor revision is addressed.

Reviewer #4: (No Response)

**Summary and General Comments**

Reviewer #1: (No Response)

Reviewer #2: I read and enjoyed the present work. The main Idea was novel, rationale and interesting, experiments were designed appropriately and the results were presented and discussed professionally. I congratulate the authors for doing this study. However, there were some minor points that I'll mention in the following.

1. Since the authors purified the recombinant proteins, I think it was better if they share some photos of the recombinant proteins.

2. The authors have done IgG isotyping, however, they did not discuss the results of this experiment in "Discussion" part.

3. Somewhere in the manuscript, mainly from line 394 to the end of the "Discussion", no reference has mentioned, though it is necessary to refer to the previous studies. For example, i) Salmonella has two type three secretion systems involved in the pathogenicity [Reference is needed], ii) pathogenicity mechanisms are different between Salmonella and Shigella and particularly in regard to the involvement of the innate and adaptative immune response, iii) a more preponderant role of the intracellular cycle of Salmonella compared to Shigella [Reference is needed] and iv) a faster systemic dissemination of Salmonella in the murine model [Reference is needed]...

Finally, in my point of view this work deserves publishing in that esteemed journal.

Best,

Reviewer #3: The manuscript is recommended for publication once the minor revision is addressed.

Reviewer #4: The goal of the manuscript was to examine the immunogenicity and protective efficacy of the homologous type-III secretion system tip proteins IpaD from Shigella flexneri and SipD from Salmonella following intranasal and intragastric administration in a mouse model. Using various ELISA assays to measure antibody responses as well as performing lethal challenge studies in mice, the authors were able to detect IgG and IgA-specific responses and demonstrated that mice immunized with the proteins protected against each pathogen. More importantly, the authors provide evidence that the proteins can provide cross-protection against the other pathogen, which has important implications for vaccine development. Throughout the study, the authors perform robust analyses with a significant number of mice per treatment group, in which mice were immunized on days 0, 21, and 42 and challenged studies were performed on day 84. Results of the antibody responses in both the figures and supplemental figures were provided for both endpoint titers and throughout the course of the immunizations. In all, the data provide promising results for the pre-clinical efficacy analysis of the antibodies generated and the protection of mice that results from immunization of IpaD and SipD proteins. However, review of the manuscript identified some major and minor concerns that are highlighted below.

Major: 

1. The authors state that the recombinant IpaD and SipD proteins were purified by cloning a poly-histidine tag on the 3’ end of the gene sequence. Moreover, the labeling of the recombinant proteins with biotin for the ELISAs appears to have the histidine tag intact for this procedure. There are no histidine-only controls provided in the analyses to determine if the antibodies generated are specific to the histidine tag. These controls are not only important for determining IpaD- or SipD-specific antibody responses, but are also important when evaluating the cross-protection analysis performed in the study. It is possible that the cross-protection is due to some or all the histidine tag present on the recombinant proteins and not due to the 38% identity between the two proteins. 

2. The authors utilize IgG ratio analyses (Figure 4) to determine if the results are indicative of a humoral or cellular immune response of T helper cells. Direct measures and identification of specific T cell responses were not performed, which would strengthen the data. In the absence of such analyses, the author should state any conclusion from the ratio analyses as a hypothesis. 

3. While the challenge studies use 100 LD50, there is concern that the LD50 dose used by the authors for S. flexneri resulted in 80% survival and not 50% survival (see supplemental figure 5). As the authors point out, the challenge dose of 5.1 x 10^10 is very high, but justify this high dose by pointing to a Shigella study in 2012 in which high LD50s were also used. This discussion is misleading, especially since the referenced study used doses of 6 x 10^7 and 1.3 x 10^8 for S. flexneri. There is a significant concern that the protection observed in the manuscript is not as promising given the dosages and model used. Furthermore, the concern is raised even more given the fact that the infection model is pulmonary-based and not physiologically relevant to Shigella infection. Animal models for Shigella can be difficult and vaccine development has not been successful despite promising animal model analyses throughout the literature. In the absence of repeating the analyses with better infection results as performed by other studies, the authors should better highlight and discuss the shortcomings from their analyses. 

Minor:

1. The figure legends should define what the controls were for the analyses. It is assumed that the controls were mice treated with adjuvant + PBS, but the clarity would be helpful. It should also be stated in the Materials and Methods how many mice were used for the control groups. 15 is assumed but clarification would be helpful. 

2. The figure legends should better clarify the immunogen and recombinant protein sentences (e.g. for Fig 1, lines 216-217). Suggestion is “…indicates biotinylated recombinant protein used for the ELISA analyses.”

3. The previous publication by the group in 2016 examined the secondary structure of recombinant SipD and PrgI using Far-UV CD spectroscopy. It is not clear why this analysis was not performed for IpaD.

PLOS authors have the option to publish the peer review history of their article (what does this mean?). If published, this will include your full peer review and any attached files.

Reviewer #1: No

Reviewer #2: Yes: Abbas Hajizade, Assistant Professor of Biotechnology

Reviewer #3: Yes: Chiuan Yee Leow

Reviewer #4: No
---

## [Decision Letter · Decision Letter 1]

26 Apr 2020

Dear Dr. Simon,

We are pleased to inform you that your manuscript 'SipD and IpaD induce a cross-protection against Shigella and Salmonella infections' has been provisionally accepted for publication in PLOS Neglected Tropical Diseases.

Best regards,

Alfredo G Torres

Deputy Editor

Alfredo Torres

Deputy Editor

Reviewer's Responses to Questions

**Key Review Criteria Required for Acceptance?**

**Methods**

-Are the objectives of the study clearly articulated with a clear testable hypothesis stated?

-Is the study design appropriate to address the stated objectives?

-Is the population clearly described and appropriate for the hypothesis being tested?

-Is the sample size sufficient to ensure adequate power to address the hypothesis being tested?

-Were correct statistical analysis used to support conclusions?

-Are there concerns about ethical or regulatory requirements being met?

Reviewer #1: (No Response)

Reviewer #2: All required questions have been answered and that all responses meet formatting specifications.

Reviewer #4: Yes

**Results**

-Does the analysis presented match the analysis plan?

-Are the results clearly and completely presented?

-Are the figures (Tables, Images) of sufficient quality for clarity?

Reviewer #1: (No Response)

Reviewer #2: All required questions have been answered and that all responses meet formatting specifications.

Reviewer #4: Yes

**Conclusions**

-Are the conclusions supported by the data presented?

-Are the limitations of analysis clearly described?

-Do the authors discuss how these data can be helpful to advance our understanding of the topic under study?

-Is public health relevance addressed?

Reviewer #1: (No Response)

Reviewer #2: All required questions have been answered and that all responses meet formatting specifications.

Reviewer #4: Yes

**Editorial and Data Presentation Modifications?**

Reviewer #1: (No Response)

Reviewer #2: Accept

Reviewer #4: (No Response)

**Summary and General Comments**

Reviewer #1: (No Response)

Reviewer #2: The manuscript is good enough to be accepted.

Reviewer #4: The authors have addressed my concerns from review of the original manuscript and have revised the manuscript accordingly. These revisions and clarifications have enhanced the quality of the manuscript.

PLOS authors have the option to publish the peer review history of their article (what does this mean?). If published, this will include your full peer review and any attached files.

Reviewer #1: No

Reviewer #2: No

Reviewer #4: No

---

## [Editor Report · Acceptance letter]

15 May 2020

Dear Dr. Simon,

We are delighted to inform you that your manuscript, "SipD and IpaD induce a cross-protection against *Shigella* and *Salmonella* infections," has been formally accepted for publication in PLOS Neglected Tropical Diseases.

Best regards,

Serap Aksoy

Editor-in-Chief

Shaden Kamhawi

Editor-in-Chief
